# A Systematic Review on Feature Extraction in Electroencephalography-Based Diagnostics and Therapy in Attention Deficit Hyperactivity Disorder

**DOI:** 10.3390/s22134934

**Published:** 2022-06-29

**Authors:** Pasquale Arpaia, Attilio Covino, Loredana Cristaldi, Mirco Frosolone, Ludovica Gargiulo, Francesca Mancino, Federico Mantile, Nicola Moccaldi

**Affiliations:** 1Department of Electrical Engineering and Information Technologies (DIETI), University of Naples “Federico II”, 80121 Naples, Italy; mirco.frosolone@unina.it (M.F.); ludovica.gargiulo@unina.it (L.G.); francesca.mancino2@unina.it (F.M.); nicola.moccaldi@unina.it (N.M.); 2Interdepartmental Research Center on Management and Innovation in Healthcare (CIRMIS), University of Naples “Federico II”, 80121 Naples, Italy; 3Villa delle Ginestre, Rehabilitation Center, 80040 Naples, Italy; attiliocovino@libero.it (A.C.); federico.mantile@fastweb.net (F.M.); 4Department of Electronics, Information e Bioengineering, Milan Polytechnic, 20133 Milan, Italy; loredana.cristaldi@polimi.it

**Keywords:** ADHD, executive function, ERP, P300, children

## Abstract

A systematic review on electroencephalographic (EEG)-based feature extraction strategies to diagnosis and therapy of attention deficit hyperactivity disorder (ADHD) in children is presented. The analysis is realized at an executive function level to improve the research of neurocorrelates of heterogeneous disorders such as ADHD. The Quality Assessment Tool for Quantitative Studies (QATQS) and field-weighted citation impact metric (Scopus) were used to assess the methodological rigor of the studies and their impact on the scientific community, respectively. One hundred and one articles, concerning the diagnostics and therapy of ADHD children aged from 8 to 14, were collected. Event-related potential components were mainly exploited for executive functions related to the cluster *inhibition*, whereas band power spectral density is the most considered EEG feature for executive functions related to the cluster *working memory*. This review identifies the most used (also by rigorous and relevant articles) EEG signal processing strategies for executive function assessment in ADHD.

## 1. Introduction

Attention deficit hyperactivity disorder (ADHD) is a neurodevelopmental disorder characterized by inattention and/or hyperactivity–impulsivity. According to the fifth edition of the Diagnostic Statistical Manual of Mental Disorders (DSM-5), inattention and/or hyperactivity–impulsivity symptoms must be present before age 12, in two or more contests, such as school and home. Impairment contributes to academic, professional, or social dysfunction. These symptoms must be present for at least 6 months and do not occur exclusively during schizophrenia or another psychotic disorder and must not be better explained by another mental disorder (mood disorder, anxiety disorder, dissociative disorder, and personality disorder). The DSM-5 lists nine symptoms related to inattention and hyperactivity–impulsivity, respectively. If the subject exhibits at least 6 of the 9 symptoms of inattention, predominantly inattentive type (ADHD-I) is diagnosed; predominantly hyperactive–impulsive type (ADHD-PH) includes at least 6 of the 9 symptoms of hyperactivity–impulsivity; the combined type (ADHD-C) involves inattention and hyperactivity–impulsivity symptoms [1].

The diagnostic criteria of DSM V are based on subjective assessments of perceived behavior. There is no mention of the use of biomarkers. Nonetheless, for 75 years, the EEG has been used for the study of ADHD [2]. EEG is recognized in the literature as one of the main candidates to provide support for the diagnosis and treatment of ADHD on a biological basis [3]. The first studies based on EEG and ADHD hypothesized a hypoarousal condition [4] of the subjects revealed by a polarisation of the EEG signal power at low frequencies. This hypothesis was confirmed by the U.S. Food and Drug Administration, which authorised the use of a device based on the ratio between power at high and low frequencies for diagnostic purposes [5]. Subsequently, further studies have not always confirmed the statistical significance of this electroencephalographic phenomenon in ADHD. Further approaches are therefore being considered for the study of such a heterogeneous phenomenon. For example, inattentive sub-types are characterized by deficits in stimuli processing speed. Therefore, ERPs assessment is particularly useful. The main advantage of ERPs is that they are able to capture the evolution of brain activity following a specific event with high temporal accuracy and thus can be used to detect sensory processing deficits [3].

In the literature, numerous studies consider the use of EEG both for diagnostics [3] and therapy [6]. To date, however, generalizable electroencephalographic patterns of ADHD have not yet been identified. According to [3], there can be no generally valid electroencephalographic features for an extremely heterogeneous phenomenon such as ADHD. DSM 5, compared with DSM 4, has increased the articulation of ADHD into subtypes. In doing so, it has also indicated a direction to go in to achieve a greater EEG-based understanding of the disorder. Anchoring EEG features to elementary cognitive functions allows the heterogeneity of ADHD-related disorders to be mapped more effectively. The correspondence between the ADHD subtype and the impairment of one or more EFs is discussed in [7]. In fact, EFs are elementary constructs that can be combined into more complex systems. The EEG features of the different executive functions can also be combined for a more targeted assessment of the different ADHD subtypes [8]. Executive functions (EF) are a set of neurocognitive processes involved in goal-oriented problem-solving [9]. According to Miyake et al., the basic EFs are *inhibition, working memory*, and *flexibility* [10]. In particular, inhibition is linked to the activation of networks involving bilateral frontal, upper right temporal occipital gyrus, and lower left, right thalamic structures and midbrain [11]. Working memory involves the dorsolateral prefrontal cortex [12], while flexibility relates to the prefrontal and posterior parietal cortex [10]. The combination of basic EF gives rise to higher-order EFs, i.e., *reasoning, problem-solving*, and *planning* [13,14,15]. In the early 2000s, Baddeley proposed the sub-articulation of the basic EFs in components (sub-functions) [16]. In particular, as an example, working memory is divided into (i) phonological loop; (ii) visuospatial sketchpad; and (iii) episodic buffer. However, a few years later, Friedman and Miyake (2004) did not consider this distinction. In recent years, an emerging trend has focused on analyzing executive sub-functions. Stahl et al. (2014) focused on two sub-EFs of inhibition, namely, *interference inhibition* and *inhibiting prepotent responses*. Rey-Mermet et al. (2017) also demonstrated how a two-component inhibition model best explained the data observed in young and older adults. A further articulation of the inhibition sub-functions was proposed by Diamond et al., in 2013. They broke down the response inhibition into *continuous response* and *response to temptations*, and they broke down interference inhibition into *selective attention* and *cognitive inhibition*.

Feature extraction is a pillar step of digital signal processing and can be summarized in (i) choosing a suitable analysis domain (e.g., time, frequency, and space); (ii) eventually, focusing on a portion of the domain (sub-domain); and (iii) applying proper mathematical functions to obtain synthetic and highly informative values of the input signals. Sometimes features extracted in this way undergo further transformation and/or calibration to improve the detection or classification process [17,18]. Currently, the relationship between EFs and EEG features is not uniquely defined. Moreover, many studies examine the EEG signal of the ADHD subject without clarifying which particular EF is being investigated. In other studies, the investigated EF is related to non-specific EEG features (i.e., those already associated with other EFs in the literature).

This review aims to improve the EEG-based approaches to the ADHD by focusing on the EEG feature of EFs. The EEG can offer a multivariate approach to the study of a heterogeneous syndrome such as ADHD, by measuring executive functions and sub-functions. Consequently, as far as EEG-based studies associating ADHD with EFs deficit are concerned, the key points of the research are reported below:Identifing EFs evaluated in EEG-based ADHD studies and their resolution level among high order-, basic-, sub-, and components of sub-EFs;Counting the articles that studied each specific relationship between an EEG feature and an EF;Reporting wether the relationships between EEG features and EFs are statistically relevant or not;Analyzing the methodological rigor of the articles and their impact within the reference scientific community.

The review is structured as follows: in Section 2, the procedures for the selection and analysis of the articles are presented. Section 3 reports the results of the quantitative and qualitative analysis. Finally, the results are discussed in Section 4.

## 2. Methods

The method consists of two steps: (i) the article selection process (Figure 1) and (ii) the article analysis (Figure 2). The article analysis identifies the relationship between EEG features and EFs and evaluates the rigor and scientific impact of the studies.

### 2.1. Article Selection Process

In this Section, the inclusion and exclusion criteria, and the database search, are presented.

#### 2.1.1. Inclusion and Exclusion Criteria

The present study was carried out in compliance with the PRISMA recommendations [19] (including the Kitchenham’s guide [20]). All the articles were selected according to the following inclusion/exclusion criteria:The age of the experimental sample: only articles recruiting six fourteen-year-old participants were included; the choice of the age range of the experimental sample was due to (i) the maturation of basic EFs; (ii) the stimulation of higher order EFs in the school environment; (iii) the greater understanding and adherence to the various tasks; and (iv) the better exclusion of other pathologies diagnosable from the age of six.The participants’ conditions during EEG signal recording: studies focused on resting state were excluded. Indeed, EFs selective activation requires specific task execution;Comorbidities: articles with the concurrent presence of other pathologies in participants were excluded to avoid these sources of interference on the EEG signals;Drug therapy: articles with participants under pharmacological treatment were excluded. Nevertheless, articles were included in case of the interruption of drug assumption at least six months before the execution of the experimental sessions. Articles were excluded if information about pharmacological therapy was not specified.The type of article: journal and conference articles were included, while reviews, commentaries, and editorials were excluded because they do not report directly on field studies.

#### 2.1.2. Database Searches

A flow diagram representation of the database search is shown in Figure 1. The identification, the screening, the eligibility, and the inclusion phases are reported. The articles were collected from Pubmed, Scopus, and IEEEXplore databases by using the query “ADHD AND EEG AND NOT Adult”. Only English results published from January 1996 to March 2021 were considered. An initial search led to 1390 articles: 689 from Pubmed, 74 from IEEEXplore, and 637 from Scopus, and 867 articles were the output of the identification phase, after removing duplicates. The full-text abstracts were analyzed in the screening phase, and 474 articles were excluded based on the selection criteria reported in Section 2.1.1. In the eligibility phase, the full text of the remaining 393 articles was screened according to the aforementioned criteria. At the end of the process, 101 articles were included: 86 from Scopus and 15 from PubMed.

### 2.2. Article Analysis Procedure

In this Section, the procedures for the executive function identification, the EEG feature identification, and the assessment of the methodological rigor and scientific impact of the articles are presented.

#### 2.2.1. Executive Function Identification

Each article was labeled by the main focused EFs. When the authors did not specify the EFs under investigation, the links between the EFs and the articles were based on the performed experimental test. In the literature and clinical practice, several tests are administered to assess the impairments of EFs. Nevertheless, an exclusive link between a test and an EF cannot be guaranteed [21,22]. However, for each test, prevalent activated EFs can be assumed in many cases [13]. Below, a description of the EFs and the main tests used for their evaluation is reported.

In ADHD studies, researchers are mainly interested in basic and sub-EF levels [8]. Inhibition is defined as the ability “to control one’s behavior, thoughts, and/or emotions to override a strong internal predisposition or external lure, and instead do what’s more appropriate or needed” [13]. This function is usually tested in an *oddball framework* [13]. The two components of inhibition, i.e., interference and response, are defined, respectively, as the ability to “filter out irrelevant information in the environment” and “ inhibit inappropriate but prepotent responses”, respectively [23]. Paradigms implemented to test these components include tasks in which subjects must ignore irrelevant stimuli [24] and inhibit prepotent response inclinations [25,26].

The working memory keeps in mind information during the execution of complex tasks [16]. This function is assessed by asking the participants to recall information previously received through multi-sensorial channels [27]. Working memory resources are separated into verbal and visuospatial constructs based on the type of information held in memory [28]. Namely, the phonological loop deals with the phonetic and phonological therapy ensuring the time properties preservation. At the same time, the visuospatial sketchpad maintains and processes the visual-spatial information and can generate mental images. The paradigms use auditory or visual contents to test these components [29,30]. Finally, cognitive flexibility represents a creative and adaptive mindset to rapid circumstance variations. In the experimental setup, the subject is required to shift among different cognitive schemes in response to a dynamic task [31]. Links among EFs and some of the main used tests are shown in Table 1.

The proposed method to identify links among EFs and articles is articulated in mutually-exclusive successive steps as follows: (i) standard tests are implemented: therefore, articles are labeled based on the test-related EFs; (ii) the article employed custom tests, but the authors clarified the investigated Efs: therefore, articles are labeled based on the declared EFs; (iii) custom tests are used, and the authors did not declare to focus on specific EFs: therefore, articles are tagged based on EFs related to the most similar standard test.

#### 2.2.2. EEG Features Identification

Once the EFs mainly investigated for each article are identified, the individuation of the link between EEG features and EFs is almost automatic. All the features collected from the articles are organized according to a multi-level pattern (Figure 3). The first level is the domain of definition: spatio-time, spatio-frequency, or spatio-time-frequency domain. In all cases, the spatial domain can be considered once the distributed mode of recording the EEG signal is given: it is acquired in a specific scalp region depending on the chosen headset.

At this level, the signal is treated by referring to peculiar pre-processing (averaging) or transformation (Fourier, Welch, and so on). As far as the second level is concerned, the sub-domains are adopted, namely, the time sub-domains and the bands (alpha, beta, theta, and so on). Finally, in third level, the identification of the features is completed by means of a synthetic value extracted after a specific operation (mean, peak, power spectral density, and so on).

In the case of articles focused on therapy, only the features subjected to experimental validation are considered.

#### 2.2.3. Assessment of the Methodological Rigor and Scientific Impact of the Articles

The quality of the collected articles was evaluated according to the PRISMA guidelines [32,33]. Namely, the Quality Assessment Tool for Quantitative Studies (QATQS) [34], a method created by researchers from Canada’s Efficient Public Health Practice Project (EPHPP), was used.

All the studies were classified according to the six components of QATQS: (1) selection bias, (2) study design, (3) confounders, (4) blinding, (5) data collection methods, and (6) withdrawal and dropouts. These components embed the criteria indicated in the Cochrane Collaboration and PRISMA declaration guidelines relating to the bias issues [19,35].

The blinding method was strictly developed in the framework of therapy effectiveness assessment, and, therefore, the blinding component was not considered for diagnostic articles.

The quality of each paper was assessed by assigning a score from 1 (high) to 3 (low) to each component. Firstly, the score was assigned by the sixth author of this review. Then, the evaluation was made by the fifth author. In case of disagreement, all the authors discussed and sought convergence.

After evaluating the rating of the components, the global rating was calculated for each article. Therefore, if no components scored 3, the article was labeled as *strong*; if only one component scored 3, the article was labeled as *moderate*; and finally, if more than one component scored 3, it was labeled as *weak*.

Finally, a further analysis based on Scopus’s *field-weighted citation impact* metric was conducted to highlight the impact of the articles on the reference scientific community. The field-weighted citation impact metric is useful to benchmark regardless of differences in disciplinary profile, age, and publication type composition, and it provides a useful way to evaluate the article’s citation performance. A value grater than 1 means that the article is more cited than the average of articles published in the same year and in the same field of interest. For example, 1.21 means 21% more cited than expected.

## 3. Results

In Table 2 EFs, EEG features, quality assessment output, and field-weighted citation impact are reported for each article. As far as quality assessment output is concerned, the normalised QATQS score (the smaller the score, the higher the quality of the article), the quality label (weak, strong, or moderate), the size of the experimental sample (N), and the use of other bio-markers (beside the EEG signal) are reported. In Section 3.1, the relationship between EEG features and EFs is focused on, while in Section 3.2, quality results are detailed.

### 3.1. Executive Functions and EEG Features

As far as the investigation level of EFs focused by the articles is concerned, the sub-EFs level is predominant (68 articles), followed by that of basic EFs (31 articles), and, lastly, that of high-order EFs (5 articles). In particular, as the sub-function level is concerned, interference inhibition is absolutely the most investigated (47 articles), with about twice as many articles on response inhibition (26 articles) and visual-spatial memory (17 articles). There are very few studies on verbal working memory (3 articles). Considering basic EFs, working memory and inhibition (16 and 13 articles, respectively) are investigated more than flexibility (3 articles). As far as high-level FEs are concerned, only planning is considered more than once (four articles) (Figure 4).

Considering that sub-functions are an articulation of basic EFs, 64 of the articles investigate the cluster *inhibition* (inhibition and its sub-functions) and the 30% the cluster *working memory* (working memory and its sub-functions). The relationship between EF clusters and EEG features was analyzed starting from their domains of definition. In particular, the feature was defined *effective* when it assumed different levels (with statistical relevance) in the group with ADHD (target group) compared to the control group (or to the pre-treatment condition if the comparison is made within the same group). Otherwise, the feature is defined as *not effective*.

The features related to the cluster inhibition are extracted at 52% in the time domain, at 45% in the frequency domain, and at 2% in the time-frequency domain. In the time domain, 13 articles consider the amplitude of P3 (See the Appendix A for details) and 9 amplitude of N1 (See the Appendix A for details) (Figure 5). In the frequency domain, 15%, 17%, and 18% analyzed the power spectrum density in alpha, beta, and theta bands, respectively (Figure 6). The WM-cluster features are mainly proposed by referring to the frequency domain (71%) (Figure 7), 28% of articles focus on time domain (Figure 8), and only 1% focus on time-frequency domain. In the frequency domain, the most considered features are the power spectrum density in alpha (16%), beta (14%), and theta (7%) bands. In the time domain, 20% of the articles considered the P3 amplitude.

As far as sub-functions are concerned, the interference inhibition features are mainly extracted from the frequency domain (65%). In particular, almost half of the articles evaluate the power spectrum density in alpha, beta, and theta bands (Figure 9). For response inhibition, the time domain is the most investigated (73%), with particular attention paid to the N1 and N2 (See the Appendix A for details) amplitudes (Figure 10). Most of the articles concerning the visuospatial WM features are centred on frequency domain (66%), but no clear trends emerged (Figure 11). As aforementioned, only three articles focus on verbal WM; therefore, any statistic on the features would be inconsistent.

### 3.2. Quality Assessment Output

The analysis results were carried out separately on the diagnostic and therapeutic articles due to the different numbers of QATQS components considered for the two type of articles, as stated in Section 2.2.3. Articles on diagnostics were grouped into 4 strong, 14 moderate, and 29 weak, following the application of the above criteria, as shown in Figure 12. Regarding the articles on therapy, 1 strong, 4 moderate, and 49 weak articles arose, as shown in Figure 13.

Moreover, each component was analyzed to evaluate some relevant trends. Regarding articles on therapy, (i) 72% of authors included a control group in addition to the target one, (ii) 9% performed a double-blinded study, (iii) 2% supported the subjective data acquired through the administration of questionnaires or with the quantitative data from other biosignal sensors besides the EEG, and (iv) 9% comprehensively described the causes of the withdrawal and dropouts.

Regarding articles on diagnostics, (i) 92% of authors included a control group in addition to the target one, (ii) 10% supported the subjective data acquired through the administration of questionnaires or with quantitative data from other biosignal sensors besides the EEG, and (iii) 23% comprehensively described the causes of the withdrawal and dropouts. Therefore, even if the most of the studies considered a control group, the prevalence of a weak score emerged due to the partial respect for the other components proposed by the QATQS.

## 4. Discussion

ADHD has been investigated at a higher level of detail in the last decades: firstly, from a clinical point of view, DSM-V identified ADHD sub-types (2013); secondly, from a scientific point of view, impaired EFs in ADHD have been studied at the level of sub-components since the early 2000s. The review results confirmed that ADHD analysis is increasingly converging on the study of the sub-EFs. Indeed, the majority of articles (65%) analyzed the sub-components of inhibition and working memory. Only 30% evaluated the basic EFs, while 5% dwelled on high-order EFs (i.e., reasoning, planning, and problem solving). In particular, the interference inhibition and visual-spatial working memory are the mainly studied sub-functions of inhibition and working memory basic EFs, respectively. A poorly studied sub-function is the verbal working memory.

Concerning cluster *inhibition*, studies are mainly centred on the EEG features extracted from time domain (53%) with respect to the frequency domain (45%) and the time-frequency domain (2%). In particular, the most investigated time-domain features are the ERP components. Several authors studied the P3 and N1 amplitudes during inhibition tasks in ADHD subjects and controls; 13 vs. 4 and 9 vs. 2 articles found a significant statistical difference for P3 and N1 amplitudes, respectively. As stated in Section 2.2.3, in the case of features from the frequency domain, the focus was on the power spectral density in the alpha, beta, and theta bands. These studies almost agreed to identify a higher alpha and beta activity in the ADHD group than in the controls group during inhibition tasks. Conversely, concerning the power spectral density in the theta bands, six articles among 21 did not confirm this EEG feature effective in discriminating or treating ADHD patients. As far as inhibition sub-functions are concerned, a slight link was found between interference inhibition and spectral density in alpha, beta, and theta bands. Among articles concerning cluster *inhibition*, ERP components were not focused only on studies on interference inhibition. The last consideration may be due to the fact that more articles on therapy fell into this category than ones on diagnostics. Indeed, ERP components are rarely used in neurofeedback due to the high latency required for their computation. In fact, by restricting the analysis to articles on diagnostics, a prevalence in considering the amplitude of the P3 component emerged.

Working memory (WM) is another EF mainly considered in ADHD. The WM-related features were evaluated at 28% in the time domain, 71% in the frequency domain, and 1% in the time-frequency domain. A trend emerged in the frequency domain between working memory and power spectral density in alpha (9 articles), beta (8 articles), and theta (11 articles) bands. Studies on working memory sub-functions focused mainly on the visuospatial component and investigated frequency EEG features, but no significant trends prevailed. The results showed that identifying clear relationships between EFs and EEG features is still challenging. However, within some fragmentation, ERP components were particularly studied. In particular, P3 amplitude emerged as the most focused EEG feature for the diagnostics and therapy of ADHD.

So in summary, power spectral density in alpha, beta, and theta bands are the most attentioned EEG features concerning interference inhibition. Instead, N1 and N2 amplitude are the most used features with regard to response inhibition. Visuospatial working memory is mainly linked to alpha and theta band power spectral density. On the other hand, studies on cognitive flexibility and verbal working memory are few and poorly convergent.

The quality of articles was analyzed to reinforce the emerged quantitative trends. The quality evaluation was conducted according to QATQS’ guidelines. The therapeutic articles on diagnostics were classified separately because the blinding component was only analyzed for the articles on therapy, as explained in Section 2.2.3. The application of QATQS criteria to the articles on diagnostics led to the identification of 9% strong articles, 30% moderate articles, and 61% weak articles, whereas it was apparent from the analysis of the articles on therapy that 2% of the articles had a strong score, 7% had a moderate score, and 91% reported a weak score. Within the moderate scoring categories, articles showing high scores on at least half of the components of the QATQS were identified as *higher quality articles*, along with articles with strong scores. In total, there are 15 higher-quality articles; 20% studied the working memory and 80% analyzed inhibition and/or its sub-functions.

As reported in Figure 14 and Figure 15, the articles focused on inhibition confirmed the effectiveness of the ERP components for diagnostics and therapy of ADHD patients. Finally, a further analysis based on Scopus’s *Field-Weighted Citation Impact* metric was conducted. Specifically, a comparison was performed focusing on the five best performing articles according to the Scopus metric (Table 3) and the five articles found to be of higher quality with the application of the QATQS (Table 4). In both the cases, the comparison between the two scores, namely, Scopus and QATQS, was realized.

In Table 3, the articles were sorted according to the field-weighted citation impact. The table includes the top five articles with the associated standardized QAQTS score.

The normalized QATQS score is computed as the ratio between the global quality score of the article and the number of quality components considered. Analyzing the five articles with the highest FWCI, only 20% of the articles report a higher quality score than the quality score median. In Table 4, the articles were sorted according to the normalized QAQTS. In particular, the five articles with the highest score are reported with the corresponding FWCI obtained. From the analysis of these articles, it emerged that 80% of the articles report a higher FWCI than the FWCI median. Finally, only one article appears in both tables, as shown in Table 3 and Table 4. However, the results obtained by means of the two metrics appear to be compatible. Indeed, four of the five articles performed better according to one criterion scored higher than the median value with respect to the other criterion.

Articles with higher scores according to the two metrics confirmed the results of the quantitative analysis. As far as the article that scored 4.17 according to Scopus parameters (highest score), visuospatial working memory is the investigated executive function. The authors identified power spectral density in theta and beta bands as the most representative features for diagnostics and therapy. The highest QATQS scored article (1.17 scored) considered both inhibition and working memory and highlighted the role of ERP components (in particular, N1, N2, and P3 latencies). Finally, the only article, highly scored according to both the metrics is focused on interference inhibition and the related EEG features are P1, P2, and N1 amplitudes. These last considerations also confirmed the centrality of the ERP components in the diagnostics and therapy of ADHD based on inhibition and the role of power spectral density for the visuospatial working memory, which already emerged from the quantitative study. This review encourages further investigation into the use of EEG in the diagnosis and therapy of ADHD based on EFs assessment.

## 5. Conclusions

A systematic review of feature extraction strategies in electroencephalographic (EEG) studies concerning the diagnosis and therapy of attention deficit hyperactivity disorder (ADHD) in children is presented. The analysis was realized at the executive function level to manage the effort of finding neurocorrelates of an heterogeneous disorders such as ADHD. One hundred and one articles, concerning the diagnostics and therapy of ADHD children aged 8 to 14, were collected. Each article was subjected to two types of analysis in parallel: (i) the analysis to extract relationships between EEG features and executive functions and (ii) the analysis to assess the rigor and scientific impact of the study (quality analysis). Event-related potential components resulted mainly exploited for executive functions related to the cluster *inhibition*, whereas band power spectral density is the most considered EEG feature for executive functions related to the cluster *working memory*. The quality analysis confirmed the quantitative results regarding the significance of the band power spectral density and event-related potential components for the analysis of executive functions in ADHD. This review identifies the most promising EEG features for the study of executive functions in ADHD. Anchoring EEG features to elementary cognitive functions allows the heterogeneity of ADHD-related disorders to be electroencephalographically analyzed more effectively.

## Figures and Tables

**Figure 1 sensors-22-04934-f001:**
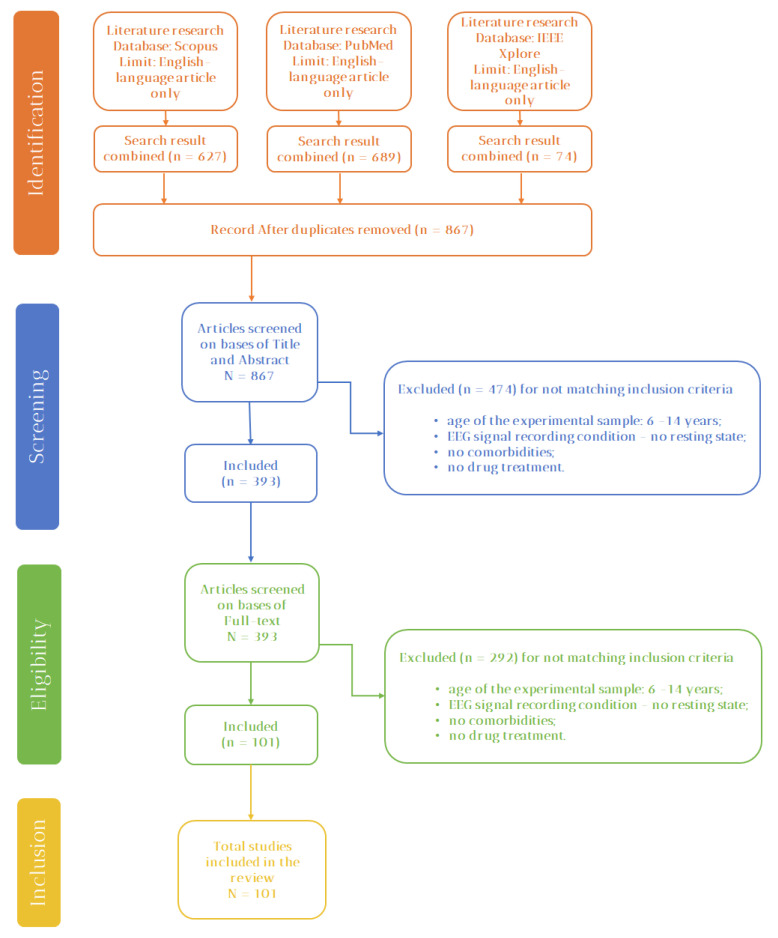
PRISMA-flow of the articles selection process.

**Figure 2 sensors-22-04934-f002:**
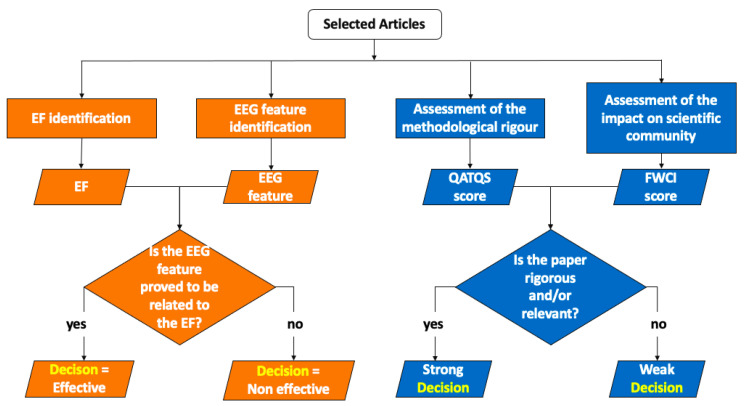
The flow-chart of the article analysis. Each article is subjected to two types of analysis in parallel: (i) the analysis to extract relationships between EEG features and executive functions (in orange) and (ii) the analysis to assess the rigor and scientific impact of the study (in blue).

**Figure 3 sensors-22-04934-f003:**
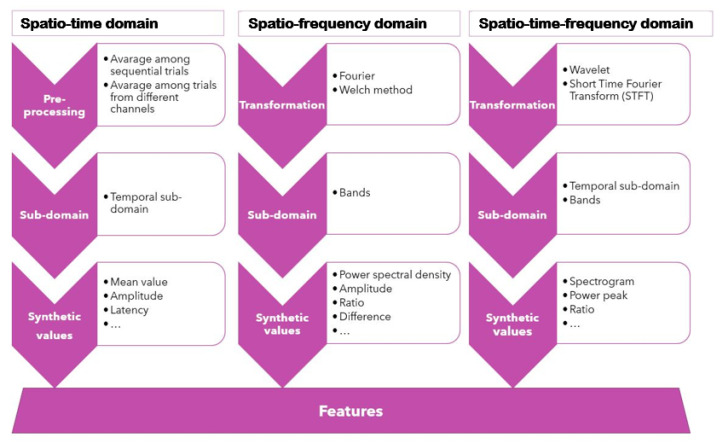
EEG features classification pattern.

**Figure 4 sensors-22-04934-f004:**
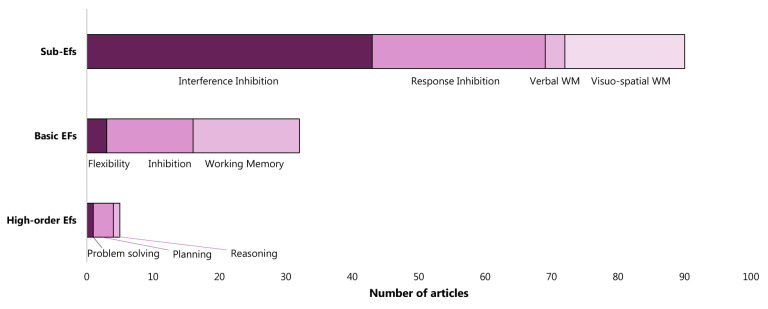
Number of articles per executive function considering the level of details in analysis of executive functions.

**Figure 5 sensors-22-04934-f005:**
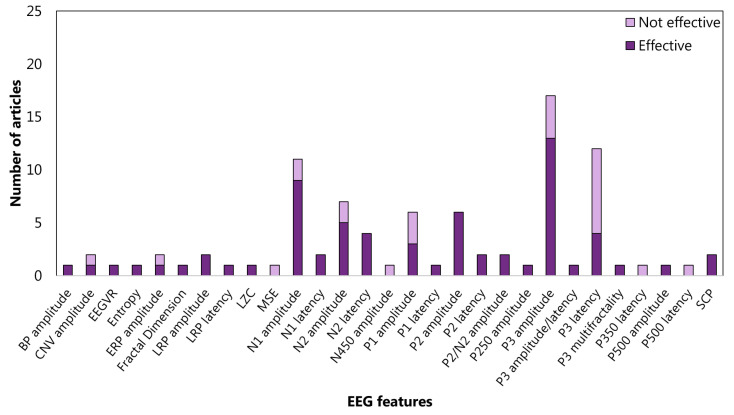
Articles focusing on the relationship between EFs from cluster *inhibition* and EEG features from time domain. P300 amplitude is the feature most studied: 12 articles verified (effective) the relationship, and 3 articles did not (not effective). LZC: Lempel–Ziv complexity; EEGVR: electroencephalogram valid rate. MSE: multi-scale entropy. SCP: slow cortical potentials; and ERP: event-related potential.

**Figure 6 sensors-22-04934-f006:**
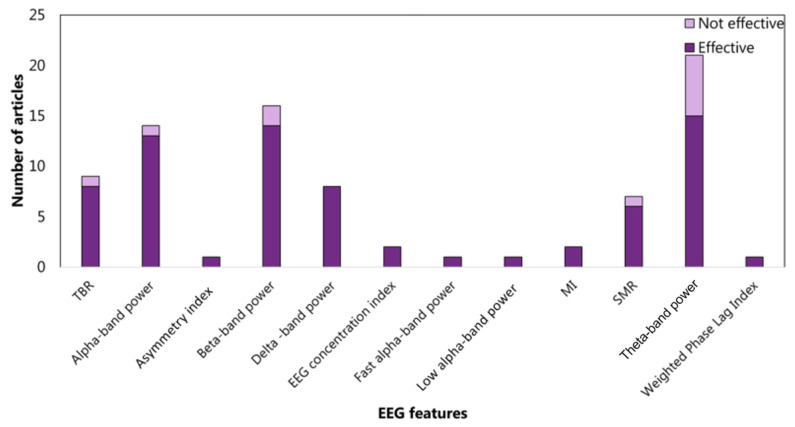
Articles focusing on the relationship between EFs from cluster *inhibition* and EEG features from frequency domain. Theta-band power is the feature most studied: 15 articles verified (effective) the relationship, and 6 articles did not (not effective). MI: modulation index. SMR: senso-motor rhythm.

**Figure 7 sensors-22-04934-f007:**
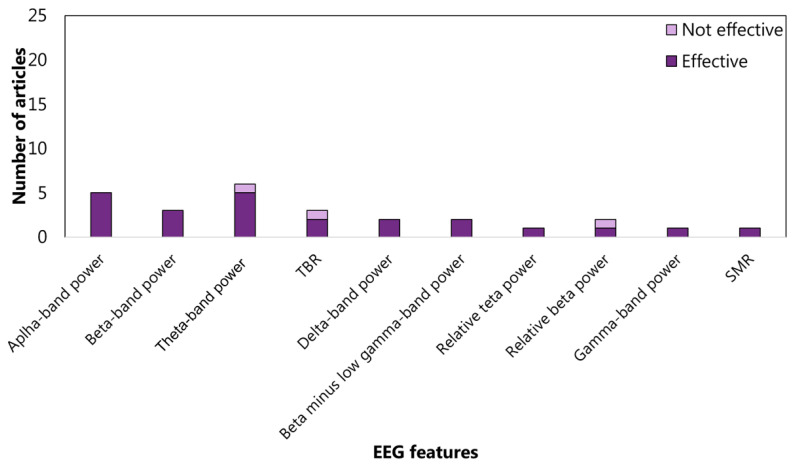
Articles focusing on the relationship between EFs from *cluster working memory* and EEG features from frequency domain. Theta-band power is the feature most studied: five articles verified (effective) the relationship. TBR: theta–beta ratio; SMR: senso-motor rhythm.

**Figure 8 sensors-22-04934-f008:**
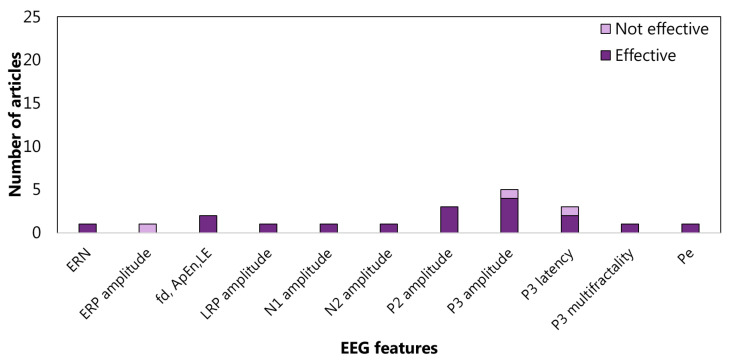
Articles focusing on the relationship between EFs from *cluster working memory* and EEG features from time domain. P300 amplitude is the feature most studied: four articles verified (effective) the relationship, and 1 article did not (was not effective). fd: fractal dimension; ApEn: Approximate Entropy; LRP: lateralised readiness potential; LE: Lyapunov Exponent; ERN: error-related negativity; and Pe: error positivity.

**Figure 9 sensors-22-04934-f009:**
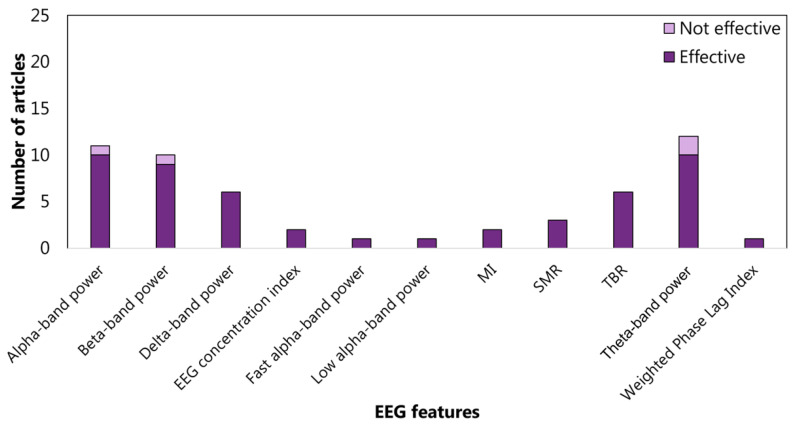
Articles focusing on the relationship between *interference inhibition* and EEG features from frequency domain. Theta-band power is the feature most studied: 10 articles verified (effective) the relationship, and 2 articles did not (not effective). TBR: theta-beta ratio; MI: modulation index; SMR: senso-motor rhythm; and CI: EEG consistency index.

**Figure 10 sensors-22-04934-f010:**
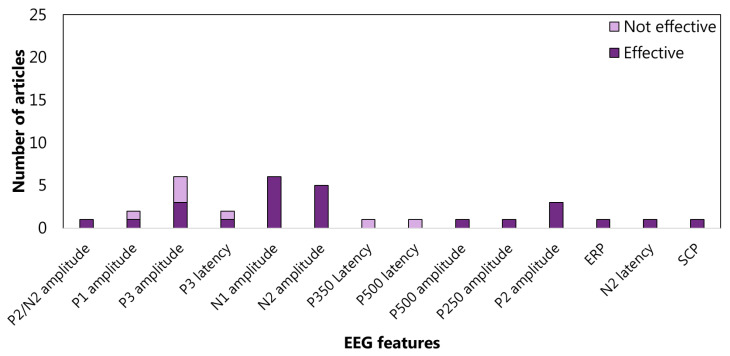
Articles focusing on the relationship between *response inhibition* and EEG features from time domain. P300 and N100 amplitudes are the features most studied. As far as P300 is concerned, three articles verified (effective) the relationship, and three articles did not (not effective), while all six articles considered verified the effectiveness of N100. ERP: event-related potential; SCP: slow cortical potentials.

**Figure 11 sensors-22-04934-f011:**
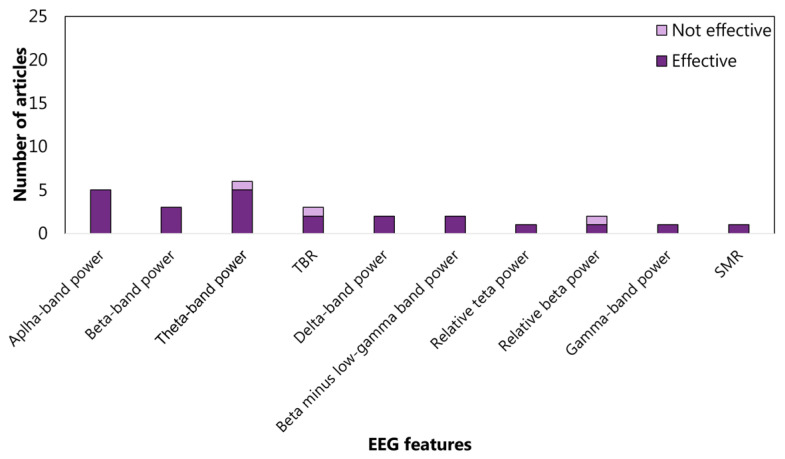
Articles focusing on the relationship between *visuo-spatial working memory* and EEG features from frequency domain. Theta-band power is the feature most studied: five articles verified (effective) the relationship, and one article did not (not effective). TBR: theta–beta ratio; SMR: senso-motor rhythm.

**Figure 12 sensors-22-04934-f012:**
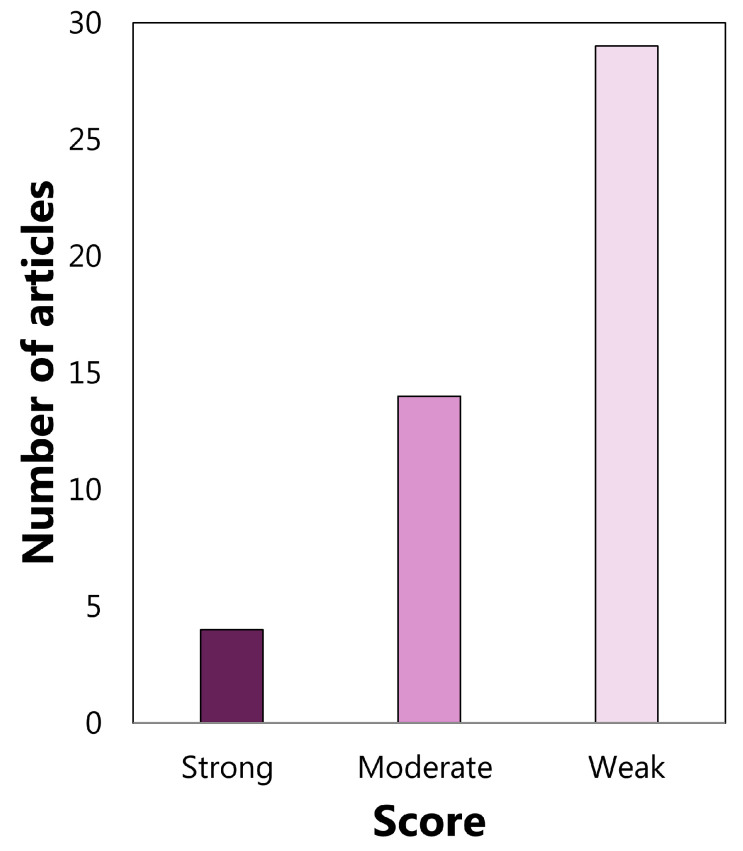
Global rating of articles on diagnostics.

**Figure 13 sensors-22-04934-f013:**
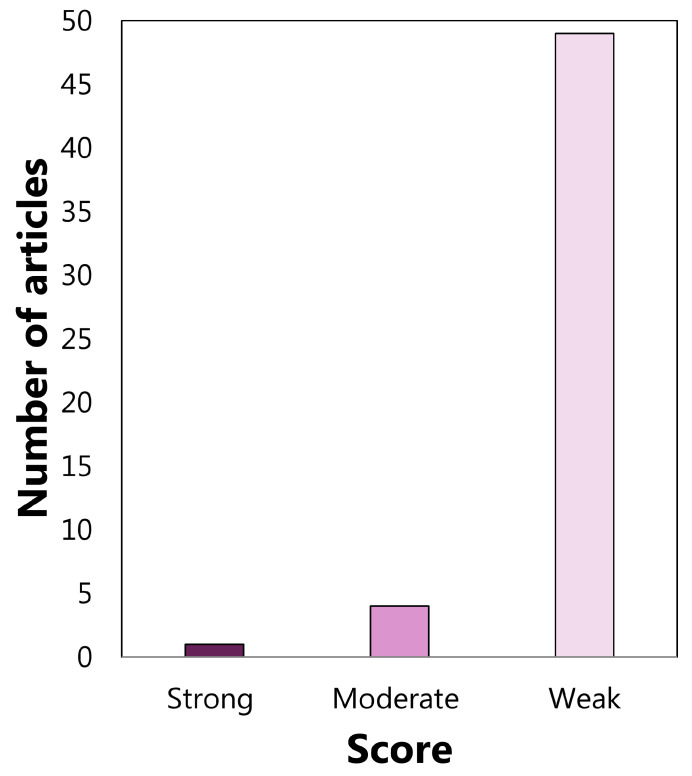
Global Rating of articles on therapy.

**Figure 14 sensors-22-04934-f014:**
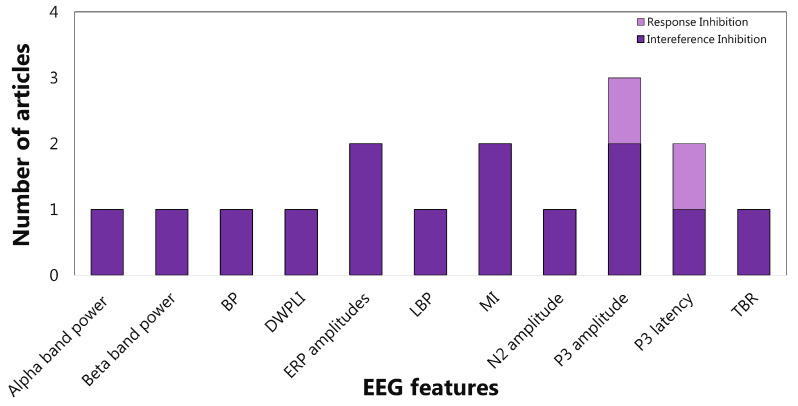
EEG features for inhibition’s sub-function emerged from highest quality articles.

**Figure 15 sensors-22-04934-f015:**
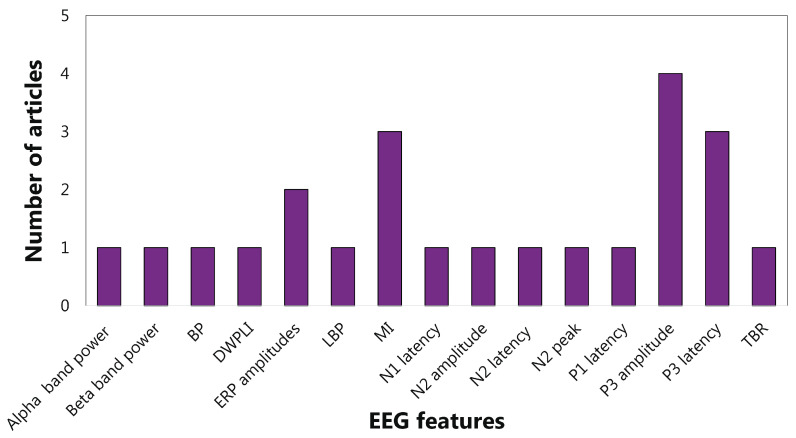
EEG features for inhibition emerged from highest quality articles.

**Table 1 sensors-22-04934-t001:** FE investigated mainly in children with ADHD and the main tests for their analysis.

Basic Executive Function	Sub-Executive Function	Main Related Test
Inhibition	Response Inhibition	Go/No Go Task [24]
	Interference Inhibition	Flanker Test [26]
Cognitive Flexibility	-	Wisconsin Card Sorting Task [31]
Working Memory	Verbal Working Memory	N-Back Task [29]
	Visual Spatial Working Memory	Corsi Block Test [30]

**Table 2 sensors-22-04934-t002:** List of articles and related executive functions. EEG features and quality scores. FWCI: field-weighted citation impact; N: the size of the experimental sample; LZC: Lempel–Ziv Complexity; EEGVR: electroencephalogram valid rate; MSE: multi-scale entropy; SCP: slow cortical potentials; TBR: theta–beta ratio; TAR: theta–alpha ratio; TBAR: theta–beta–alpha ratio; SMR: sensomotor rhythm; WPLI: weighted phase lag tndex; ERN: error-related negativity; FD: fractal dimension; BP: bereitschaftspotential; LRP: lateralised readiness potential; CNV: contingent negative variation; ERP: event-related potential; PSD: power spectral density; MI: modulation index; CI: consistency index; ERD: event-related desynchronization; ERS: event-related synchronization; and ITC: inter-trial coherence. See the Appendix for more information about some of the EEG features considered. ERP components are evaluated in terms of amplitude or latency. n.a.: not available.

Articles	Authors	Executive Functions	EEG-Features	Quality Assessment	FWCI
Computer-based inhibitory control training in children with Attention-Deficit/Hyperactivity Disorder (ADHD): evidence for behavioral and neural impact [36]	Meyer, K.N.; Santillana, R.; Miller, B.; Clapp, W.; Way, M.; Bridgman-Goines, K.; Sheridan, M.A.	inhibition	ERP-N2	Global QATQS Rating: moderate (score = 1.67); No additional bio-markers; N = 40	0.41
Executive dysfunction in medication-naïve children with ADHD: a multi-modal fNIRS and EEG study [37]	Kaga, Y.; Ueda, R.; Tanaka, M.; Kita, Y.; Suzuki, K.; Okumura, Y.; Egashira, Y.; Shirakawa, Y.; Mitsuhashi, S.; Kitamura, Y.; et al.	inhibition	ERP-N2, ERP-P3	Global QATQS Rating: moderate (score = 1.60); Additional bio-markers; N = 20	0.93
Effect of combined neurofeedback and game-based cognitive training on the treatment of ADHD: a randomized controlled study [38]	Rajabi, S.; Pakize, A.; Moradi, N.	working memory	PSD-TBR, PSD-SMR	Global QATQS Rating: strong (score = 1.67); No additional bio-markers; N = 32	0.58
Individualized neurofeedback training may help achieve long-term improvement of working memory in children with ADHD [39]	Dobrakowski, P.; Łebecka, G.	working memory	PSD-Beta, PSD-Theta	Global QATQS Rating: strong (score = 1.83); No additional bio-markers; N = 48	4.17
Increased mirror overflow movements in ADHD are associated with altered EEG alpha/beta band desynchronization [40]	McAuliffe, D.; Hirabayashi, K.; Adamek, J.H.; Luo, Y.; Crocetti, D.; Pillai, A.S.; Zhao, Y.; Crone, N.E.; Mostofsky, S.H.; Ewen, J.B.	inhibition	PSD-Alpha, PSD-Beta	Global QATQS Rating: strong (score = 2.00); No additional bio-markers; N = 50	0.77
Event-related potentials (ERPs) and other EEG based methods for extracting biomarkers of brain dysfunction: examples from pediatric attention deficit/hyperactivity disorder (ADHD) [41]	Ogrim, G.; Kropotov, J.D.	inhibition	ERP	Global QATQS Rating: strong (score = 2.20); No additional bio-markers; N = 128	0.00
Alpha modulation during working memory encoding predicts neurocognitive impairment in ADHD [42]	Lenartowicz, A.; Truong, H.; Salgari, G.C.; Bilder, R.M.; McGough, J.; McCracken, J.T.; Loo, S.K.	working memory	PSD-alpha, PSD-Theta, ERP-P2	Global QATQS Rating: moderate (score = 1.40); No additional bio-markers; N = 119	2.24
Bereitschaftspotential and lateralized readiness potential in children with attention deficit hyperactivity disorder: altered motor system activation and effects of methylphenidate [43]	Jarczok, T.A.; Haase, R.; Bluschke, A.; Thiemann, U.; Bender, S.	inhibition	LRP, BP	Global QATQS Rating: moderate (score = 1.67); Additional bio-markers; N = 33	0.55
Oscillatory neural networks underlying resting-state, attentional control and social cognition task conditions in children with ASD, ADHD and ASD+ ADHD [44]	Shephard, E.; Tye, C.; Ashwood, K.L.; Azadi, B.; Johnson, M.H.; Charman, T.; Asherson, P.; McLoughlin, G.; Bolton, P.F.	inhibition, working memory	CNV	Global QATQS Rating: moderate (score = 1.40); No additional bio-markers; N = 92	0.41
Evidence for an altered architecture and a hierarchical modulation of inhibitory control processes in ADHD [45]	Chmielewski, W.; Bluschke, A.; Bodmer, B.; Wolff, N.; Roessner, V.; Beste, C.	inhibition	ERP-P3, ERP-N2	Global QATQS Rating: moderate (score = 1.40); Additional bio-markers; N = 50	0.80
Diagnosis of attention deficit hyperactivity disorder with combined time and frequency features [46]	Altınkaynak, M.; Dolu, N.; Güven, A.; Pektaş, F.; Özmen, S.; Demirci, E.; İzzetoğlu, M.	inhibition	ERP-P3	Global QATQS Rating: moderate (score = 1.80); Additional bio-markers; N = 46	0.76
Lateral prefrontal theta oscillations reflect proactive cognitive control impairment in males with attention deficit hyperactivity disorder [47]	Zamorano, F.; Kausel, L.; Albornoz, C.; Lavin, C.; Figueroa-Vargas, A.; Stecher, X.; Aragón-Caqueo, D.; Carrasco, X.; Aboitiz, F.; Billeke, P.	inhibition	ERP-P3	Global QATQS Rating: strong (score = 2.00); No additional bio-markers; N = 54	0.42
Refining the picture of reduced alerting responses in ADHD–A single-trial analysis of event-related potentials [48]	Heinrich, H.; Busch, K.; Studer, P.; Erbe, K.; Moll, G.H.; Kratz, O.	inhibition	ERP-P3	Global QATQS Rating: strong (score = 2.20); No additional bio-markers; N = 43	0.21
A brain–computer interface based attention training program for treating attention deficit hyperactivity disorder [49]	Lim, C.G.; Lee, T.S.; Guan, C.; Fung, D.S.S.; Zhao, Y.; Teng, S.S.W.; Zhang, H.; Krishnan, K.R.R.	inhibition	PSD-sum of all bands	Global QATQS Rating: strong (score = 1.80); No additional bio-markers; N = 20	1.21
Frontal alpha asymmetry predicts inhibitory processing in youth with attention deficit/hyperactivity disorder [50]	Ellis, A.J.; Kinzel, C.; Salgari, G.C.; Loo, S.K.	inhibition	ERP	Global QATQS Rating: moderate (score = 1.80); No additional bio-markers; N = 50	0.58
Different cortical source activation patterns in children with attention deficit hyperactivity disorder during a time reproduction task [51]	Khoshnoud, S.; Shamsi, M.; Nazari, M.A.; Makeig, S.	working memory, inhibition	CNV, ERP-P3, ERP-P5	Global QATQS Rating: strong (score = 2.00); No additional bio-markers; N = 34	1.21
On the efficiency of individualized theta/beta ratio neurofeedback combined with forehead EMG training in ADHD children [52]	Bazanova, O.M.; Auer, T.; Sapina, E.A.	inhibition	PSD-beta, PSD-theta	Global QATQS Rating: strong (score = 2.00); Additional bio-markers; N = 117	1.64
Complexity analysis of brain activity in attention-deficit/hyperactivity disorder: a multiscale entropy analysis [53]	Chenxi, L.; Chen, Y.; Li, Y.; Wang, J.; Liu, T.	inhibition	PSD-beta, PSD-theta, PSD-alpha, PSD-beta, PSD-delta, MSE	Global QATQS Rating: strong (score = 2.40); No additional bio-markers; N = 26	0.78
A randomized controlled trial into the effects of neurofeedback, methylphenidate, and physical activity on EEG power spectra in children with ADHD [54]	Janssen, T.W.; Bink, M.; Geladé, K.; van Mourik, R.; Maras, A.; Oosterlaan, J.	inhibition	PSD-TBR	Global QATQS Rating: moderate (score = 1.67); No additional bio-markers; N = 112	2.75
Effect of EEG biofeedback on cognitive flexibility in children with attention deficit hyperactivity disorder with and without epilepsy [55]	Bakhtadze, S.; Beridze, M.; Geladze, N.; Khachapuridze, N.; Bornstein, N.	flexibility	PSD-SMR, PSD-beta, PSD-gamma	Global QATQS Rating: strong (score = 2.17); No additional bio-markers; N = 69	0.76
Electroencephalogram complexity analysis in children with attention-deficit/hyperactivity disorder during a visual cognitive task [56]	Zarafshan, H.; Khaleghi, A.; Mohammadi, M.R.; Moeini, M.; Malmir, N.	working memory	LZC	Global QATQS Rating: strong (score = 2.20); No additional bio-markers; N = 64	0.58
Electroencephalogram valid rate in simple reaction time task as an easy index of children’s attention functions [57]	Liao, Y.C.; Guo, N.W.; Lei, S.H.; Fang, J.H.; Chen, J.J.; Su, B.Y.; Chen, S.J.; Tsai, H.F.	inhibition	EEGVR	Global QATQS Rating: strong (score = 2.20); No additional bio-markers; N = 50	0.17
Development and evaluation of an interactive electroencephalogram-based neurofeedback system for training attention and attention defects in children [58]	Israsena, P.; Hemrungrojn, S.; Sukwattanasinit, N.; Maes, M.	reasoning	PSD-beta/a, lpha ratio	Global QATQS Rating: strong (score = 2.60); No additional bio-markers; N = 28	0.11
Use of EEG beta-1 power and theta/beta ratio over Broca’s area to confirm diagnosis of attention deficit/hyperactivity disorder in children [59]	Sangal, R.B.; Sangal, J.M.	inhibition	PSD-beta, PSD-theta, PSD-TBR	Global QATQS Rating: strong (score = 1.83); No additional bio-markers; N = 86	0.68
Neurofeedback, pharmacological treatment and behavioral therapy in hyperactivity: multilevel analysis of treatment effects on electroencephalography [60]	Moreno-García, I.; Delgado-Pardo, G.; De Rey, C.C.V.; Meneres-Sancho, S.; Servera-Barceló, M.	inhibition. working memory	PSD-beta, PSD-theta	Global QATQS Rating: strong (score = 2.00); No additional bio-markers; N = 57	1.06
Neurofeedback training intervention for enhancing working memory function in attention deficit and hyperactivity disorder (ADHD) Chinese students [61]	Wang, Z.	working memory	PSD-alpha	Global QATQS Rating: strong (score = 2.33); No additional bio-markers; N = 24	0.77
EEG differences in ADHD-combined type during baseline and cognitive tasks [62]	Swartwood, J.N.; Swartwood, M.O.; Lubar, J.F.; Timmermann, D.L.	working memory, inhibition, planning, problem solving	PSD-beta, PSD-alpha, PSD-theta, PSD-delta	Global QATQS Rating: strong (score = 2.33); No additional bio-markers; N = 56	0.39
Children with ADHD shown different alpha, beta and SMR EEG bands during habil motor tasks with high attention demand [63]	Silva, V.F.d.; Calomeni, M.R.; Borges, C.J.; Militão, A.G.; Freire, I.d.A.; Simões, K.M.; Arêas, N.T.; Silva, P.B.d.; Cabral, P.U.L.; Valentim-Silva, J.R.	flexibility	PSD-beta, PSD-alpha, PSD-SMR	Global QATQS Rating: strong (score = 2.67); No additional bio-markers; N = 14	0.00
Frequency bands in seeing and remembering: comparing ADHD and typically developing children [64]	Fabio, R.A.; Tindara, C.; Nasrin, M.; Antonio, G.; Gagliano, A.; Gabriella, M.	working memory	PSD-beta, PSD-alpha, PSD-theta	Global QATQS Rating: strong (score = 2.20); Additional bio-markers; N = 46	3.25
Decision support algorithm for diagnosis of ADHD using electroencephalograms [65]	Abibullaev, B.; An, J.	working memory	PSD-alpha, PSD-theta, PSD-theta, PSD-theta/alpha ratio, PSD-TBR, PSD-relative delta, PSD-relative beta	Global QATQS Rating: strong (score = 2.40); No additional bio-markers; N = 10	0.89
Dynamic changes in quantitative electroencephalogram during continuous performance test in children with attention-deficit/hyperactivity disorder [66]	Nazari, M.A.; Wallois, F.; Aarabi, A.; Berquin, P.	working memory	PSD-relative beta, PSD-relative alpha, PSD-relative theta, PSD-relative delta, PSD-relative TBR	Global QATQS Rating: strong (score = 2.17); No additional bio-markers; N = 32	1.26
Designing a brain-computer interface device for neurofeedback using virtual environments [67]	Yan, N.; Wang, J.; Liu, M.; Zong, L.; Jiao, Y.; Yue, J.; Lv, Y.; Yang, Q.; Lan, H.; Liu, Z.	working memory	PSD-relative TBR, PSD-relative SMR	Global QATQS Rating: strong (score = 3.00); No additional bio-markers; N = 12	0.27
Changes in cognitive evoked potentials during non pharmacological treatment in children with attention deficit/hyperactivity disorder [68]	Bakhtadze, S.; Dzhanelidze, M.; Khachapuridze, N.	inhibition	PSD-relative TBR, PSD-relative SMR, PSD-alpha	Global QATQS Rating: strong (score = 2.33); No additional bio-markers; N = 93	0.51
EEG spectral analysis of attention in ADHD: implications for neurofeedback training? [69]	Heinrich, H.; Busch, K.; Studer, P.; Erbe, K.; Moll, G.H.; Kratz, O.	inhibition	PSD-alpha, PSD-beta, PSD-theta	Global QATQS Rating: strong (score = 2.17); No additional bio-markers; N = 43	1.79
The effects of individual upper alpha neurofeedback in ADHD: an open-label pilot study [70]	Escolano, C.; Navarro-Gil, M.; Garcia-Campayo, J.; Congedo, M.; Minguez, J.	inhibition	PSD-alpha	Global QATQS Rating: strong (score = 2.50); No additional bio-markers; N = 17	1.20
A proposed multisite double-blind randomized clinical trial of neurofeedback for ADHD: need, rationale, and strategy [71]	Kerson, C.; Group, C.N.	inhibition, working memory	PSD-TBR	Global QATQS Rating: strong (score = 2.33); No additional bio-markers; N = 180	2.41
Functional disconnection of frontal cortex and visual cortex in attention-deficit/hyperactivity disorder [72]	Mazaheri, A.; Coffey-Corina, S.; Mangun, G.R.; Bekker, E.M.; Berry, A.S.; Corbett, B.A.	inhibition	PSD-alpha, PSD- theta	Global QATQS Rating: strong (score = 2.50); No additional bio-markers; N = 25	2.29
Quantative EEG during baseline and various cognitive tasks in children with attention deficit/hyperactivity disorder [73]	Bakhtadze, S.; Janelidze, M.	inhibition, working memory	PSD-alpha, PSD-delta	Global QATQS Rating: strong (score = 2.33); No additional bio-markers; N = 32	0.26
Frontal theta/beta ratio changes during TOVA in Egyptian ADHD children [74].	Halawa, I.F.; El Sayed, B.B.; Amin, O.R.; Meguid, N.A.; Kader, A.A.A.	inhibition	PSD-TBR	Global QATQS Rating: strong (score = 2.20); No additional bio-markers; N = 104	0.21
Desynchronization of theta-phase gamma-amplitude coupling during a mental arithmetic task in children with attention deficit/hyperactivity disorder [75]	Kim, J.W.; Kim, B.N.; Lee, J.; Na, C.; Kee, B.S.; Min, K.J.; Han, D.H.; Kim, J.I.; Lee, Y.S.	working memory	PSD-alpha-PSD-delta, PSD- theta, PSD- synchronization index (SI))-theta-gamma	Global QATQS Rating: strong (score = 1.80); No additional bio-markers; N = 97	0.66
Near-infrared spectroscopy (NIRS) neurofeedback as a treatment for children with attention deficit hyperactivity disorder (ADHD)—a pilot study [76]	Marx, A.M.; Ehlis, A.C.; Furdea, A.; Holtmann, M.; Banaschewski, T.; Brandeis, D.; Rothenberger, A.; Gevensleben, H.; Freitag, C.M.; Fuchsenberger, Y.; et al.	inhibition, flexibility	SCP	Global QATQS Rating: strong (score = 2.17); Additional bio-markers; N = 27	2.73
Children with ADHD show impairments in multiple stages of information processing in a Stroop task: an ERP study [77]	Kóbor, A.; Takács, Á.; Bryce, D.; Szucs, D.; Honbolygó, F.; Nagy, P.; Csépe, V.	inhibition	ERP-N1, ERP-P1, ERP-N450, LRP, SCP	Global QATQS Rating: strong (score = 2.00); Additional bio-markers; N = 24	0.61
Increased reaction time variability in attention-deficit hyperactivity disorder as a response-related phenomenon: evidence from single-trial event-related potentials [78]	Saville, C.W.; Feige, B.; Kluckert, C.; Bender, S.; Biscaldi, M.; Berger, A.; Fleischhaker, C.; Henighausen, K.; Klein, C.	working memory	LRP, ERP-P3	Global QATQS Rating: strong (score = 2.00); No additional bio-markers; N = 45	2.70
Small-world brain functional networks in children with attention-deficit/hyperactivity disorder revealed by EEG synchrony [79]	Liu, T.; Chen, Y.; Lin, P.; Wang, J.	inhibition	cluster coefficient C, and characteristic path length L-alpha, beta, theta, delta	Global QATQS Rating: strong (score = 2.40); No additional bio-markers; N = 26	1.56
Electroencephalography correlates of spatial working memory deficits in attention-deficit/hyperactivity disorder: vigilance, encoding, and maintenance [80]	Lenartowicz, A.; Delorme, A.; Walshaw, P.D.; Cho, A.L.; Bilder, R.M.; McGough, J.J.; McCracken, J.T.; Makeig, S.; Loo, S.K.	working memory	ERP-P2, PSD-TBR	Global QATQS Rating: strong (score = 2.00); No additional bio-markers; N = 99	2.66
First clinical trial of tomographic neurofeedback in attention-deficit/hyperactivity disorder: evaluation of voluntary cortical control [81]	Liechti, M.D.; Maurizio, S.; Heinrich, H.; Jäncke, L.; Meier, L.; Steinhausen, H.C.;Walitza, S.; Drechsler, R.; Brandeis, D.	inhibition	SCP	Global QATQS Rating: strong (score = 2.00); No additional bio-markers; N = 13	2.12
Visual sensory processing deficit in the occipital region in children with attention-deficit/hyperactivity disorder as revealed by event-related potentials during cued continuous performance test [82]	Nazari, M.; Berquin, P.; Missonnier, P.; Aarabi, A.; Debatisse, D.; De Broca, A.; Wallois, F.	inhibition	ERP-N2, ERP-P1	Global QATQS Rating: strong (score = 2.50); No additional bio-markers; N = 30	0.72
Do children with ADHD and/or PDD-NOS differ in reactivity of alpha/theta ERD/ERS to manipulations of cognitive load and stimulus relevance? [83]	Gomarus, H.K.; Wijers, A.A.; Minderaa, R.B.; Althaus, M.	working memory, inhibition	ERS, ERD	Global QATQS Rating: strong (score = 2.17); No additional bio-markers; N = 60	0.09
Slow cortical potential neurofeedback in attention deficit hyperactivity disorder: is there neurophysiological evidence for specific effects? [84]	Doehnert, M.; Brandeis, D.; Straub, M.; Steinhausen, H.C.; Drechsler, R.	inhibition	CNV	Global QATQS Rating: strong (score = 2.50); No additional bio-markers; N = 26	2.13
Longitudinal change of ERP during cued continuous performance test in child with attention-deficit/hyperactivity disorder [85]	Okazaki, S.; Ozaki, H.; Maekawa, H.; Futakami, S.	inhibition	ERP-P3, ERP-N2, ERP-P2	Global QATQS Rating: strong (score = 2.50); No additional bio-markers; N = 1	0.00
Exogenous orienting of visual-spatial attention in ADHD children [86]	Ortega, R.; López, V.; Carrasco, X.; Anllo-Vento, L.; Aboitiz, F.	inhibition	ERP-N1, ERP-P3, ERP-P2, ERP-CNV	Global QATQS Rating: strong (score = 2.17); No additional bio-markers; N = 60	0.92
EEG classification of ADHD and normal children using non-linear features and neural network [87]	Mohammadi, M.R.; Khaleghi, A.; Nasrabadi, A.M.; Rafieivand, S.; Begol, M.; Zarafshan, H.	working memory	ApEn, LE, FD	Global QATQS Rating: strong (score = 2.17); No additional bio-markers; N = 60	3.06
Diagnose ADHD disorder in children using convolutional neural network based on continuous mental task EEG [88]	Moghaddari, M.; Lighvan, M.Z.; Danishvar, S.	working memory	Amplitude of alpha, theta, beta+low gamma frequency bands	Global QATQS Rating: strong (score = 2.50); No additional bio-markers; N = 61	0.64
Combining functional near-infrared spectroscopy and EEG measurements for the diagnosis of attention-deficit hyperactivity disorder [89]	Güven, A.; Altınkaynak, M.; Dolu, N.; İzzetoğlu, M.; Pektaş, F.; Özmen, S.; Demirci, E.; Batbat, T.	inhibition	ERP-P3, LZC, FD	Global QATQS Rating: weak (score = 1.60); Additional bio-markers; N = 44	1.20
Methodology proposal of ADHD classification of children based on cross recurrence plots [90]	Aceves-Fernandez, M.	working memory	Recurrence rate, Determinism, Entropy, Laminarity, Trapping Time, Trend	Global QATQS Rating: strong (score = 2.20); No additional bio-markers; N = 121	0.00
Improved neuronal regulation in ADHD: an application of 15 sessions of photic-driven EEG neurotherapy [91]	Patrick, G.J.	inhibition	PSD-theta, PSD-SMR	Global QATQS Rating: strong (score = 2.67); No additional bio-markers; N = 25	0.57
Quantitative EEG differences in a nonclinical sample of children with ADHD and undifferentiated ADD [92]	M. A. Nazari, F. Wallois, A. Aarabi, P. Berquin,	inhibition	ERP-N1, ERP-P1, ERP-P3	Global QATQS Rating: strong (score = 2.00); No additional bio-markers; N = 32	4.16
Neuroelectric mapping reveals precursor of stop failures in children with attention deficits [93]	Brandeis, D.; van Leeuwen, T.H.; Rubia, K.; Vitacco, D.; Steger, J.; Pascual-Marqui, R.D.; Steinhausen, H.C.	inhibition	ERP-N1, ERP-P1, ERP-P2, ERP-P460, ERP-P550, ERP-P640	Global QATQS Rating: strong (score = 2.00); No additional bio-markers; N = 15	3.76
Electroencephalographic and psychometric differences between boys with and without attention-deficit/hyperactivity disorder (ADHD): a pilot study [94]	Cox, D.J.; Kovatchev, B.P.; Morris, J.B.; Phillips, C.; Hill, R.J.; Merkel, L.	inhibition	PSD-theta, PSD-alpha, PSD-theta	Global QATQS Rating: strong (score = 2.33); No additional bio-markers; N = 8	1.06
Audio-visual entrainment program as a treatment for behavior disorders in a school setting [95]	Joyce, M.; Siever, D.	inhibition	PSD-alpha, PSD-beta, PSD-SMR	Global QATQS Rating: strong (score = 2.67); No additional bio-markers; N = 34	0.21
EEG biofeedback training and attention-deficit/hyperactivity disorder in an elementary school setting [96]	Carmody, D.P.; Radvanski, D.C.; Wadhwani, S.; Sabo, M.J.; Vergara, L.	inhibition	PSD-beta, PSD-delta, PSD-SMR	Global QATQS Rating: strong (score = 2.67); No additional bio-markers; N = 16	0.00
A psychophysiological marker of attention deficit/hyperactivity disorder (ADHD)—defining the EEG consistency index [97]	Kovatchev, B.; Cox, D.; Hill, R.; Reeve, R.; Robeva, R.; Loboschefski, T.	inhibition	CI	Global QATQS Rating: strong (score = 2.20); No additional bio-markers; N = 35	0.61
A potential electroencephalography and cognitive biosignature for the child behavior checklist–dysregulation profile [98]	McGough, J.J.; McCracken, J.T.; Cho, A.L.; Castelo, E.; Sturm, A.; Cowen, J.; Piacentini, J.; Loo, S.K.	inhibition	PSD-alpha, PSD-beta, PSD-theta, PSD-delta	Global QATQS Rating: strong (score = 2.00); No additional bio-markers; N = 2	0.36
The effects of neurofeedback training on concentration in children with attention deficit/hyperactivity disorder [99]	Kim, S.K.; Yoo, E.Y.; Lee, J.S.; Jung, M.Y.; Park, S.H.; Park, J.H.	inhibition	EEG concentration index	Global QATQS Rating: strong (score = 2.83); No additional bio-markers; N = 3	0.31
EEG dynamics of a go/nogo task in children with ADHD [100]	Baijot, S.; Cevallos, C.; Zarka, D.; Leroy, A.; Slama, H.; Colin, C.; Deconinck, N.; Dan, B.; Cheron, G.	inhibition	ERP, ITC	Global QATQS Rating: strong (score = 2.00); No additional bio-markers; N = 14	0.68
Electroencephalographic activity before and after cognitive effort in children with attention deficit/hyperactivity disorder [101]	Buyck, I.; Wiersema, J.R.	working memory	PSD-alpha, PSD-beta, PSD-theta, PSD-TBR	Global QATQS Rating: strong (score = 2.33); No additional bio-markers; N = 43	0.48
A randomized controlled trial of a brain-computer interface based attention training program for ADHD [102]	Lim, C.G.; Poh, X.W.W.; Fung, S.S.D.; Guan, C.; Bautista, D.; Cheung, Y.B.; Zhang, H.; Yeo, S.N.; Krishnan, R.; Lee, T.S.Buchmann, J.; Gierow, W.; Reis, O.; Haessler, F.	inhibition	PSD-alpha, PSD-beta, PSD-theta,	Global QATQS Rating: strong (score = 2.17); No additional bio-markers; N = 172	1.27
Intelligence moderates impulsivity and attention in ADHD children: an ERP study using a go/nogo paradigm [103]	Buchmann, J.; Gierow, W.; Reis, O.; Haessler, F.	inhibition	ERP-P3	Global QATQS Rating: strong (score = 2.40); No additional bio-markers; N = 15	0.90
Motor cortical inhibition in ADHD: modulation of the transcranial magnetic stimulation-evoked N100 in a response control task [104]	D’Agati, E.; Hoegl, T.; Dippel, G.; Curatolo, P.; Bender, S.; Kratz, O.; Moll, G.H.; Heinrich, H.	inhibition	ERP-N1	Global QATQS Rating: strong (score = 2.17); No additional bio-markers; N = 37	0.85
ERP correlates of selective attention and working memory 654 capacities in children with ADHD and/or PDD-NOS [105]	Gomarus, H.K.; Wijers, A.A.; Minderaa, R.B.; Althaus, M.	inhibition, working memory	ERP	Global QATQS Rating: moderate (score = 1.60); No additional bio-markers; N = 60	0.59
Error and feedback processing in children with ADHD and children with autistic spectrum disorder: an EEG event-related potential study [106]	Groen, Y.; Wijers, A.A.; Mulder, L.J.; Waggeveld, B.; Minderaa, R.B.; Althaus, M.	working memory	ERP-P3, ERP-P2, ERP-Pe	Global QATQS Rating: moderate (score = 2.00); No additional bio-markers; N = 72	2.33
Changes in EEG spectrograms, event-related potentials, and event-related desynchronization induced by relative beta training in ADHD children [107]	Kropotov, J.D.; Grin-Yatsenko, V.A.; Ponomarev, V.A.; Chutko, L.S.; Yakovenko, E.A.; Nikishena, I.S.	inhibition	PSD-relative beta	Global QATQS Rating: strong (score = 2.50); No additional bio-markers; N = 86	0.43
Functional connectivity of frontal cortex in healthy and ADHD children reflected in EEG coherence [108]	Murias, M.; Swanson, J.M.; Srinivasan, R.	working memory	PSD, Coherence Power	Global QATQS Rating: strong (score = 2.33); No additional bio-markers; N = 63	1.46
Case study: improvements in IQ score and maintenance of gains following EEG biofeedback with mildly developmentally delayed twins [109]	Fleischman, M.J.; Othmer, S.	inhibition	PSD-SMR	Global QATQS Rating: strong (score = 2.50); No additional bio-markers; N = 2	0.00
A controlled study of the effectiveness of EEG biofeedback training on children with attention deficit hyperactivity disorder [110]	Zhonggui, X.; Shuhua, S.; Haiqing, X.	working memory, inhibition	PSD-SMR, PSD-theta	Global QATQS Rating: strong (score = 2.50); No additional bio-markers; N = 60	0.10
ERPs correlates of EEG relative beta training in ADHD children [111]	Kropotov, J.D.; Grin-Yatsenko, V.A.; Ponomarev, V.A.; Chutko, L.S.; Yakoveuhua, E.A.; Nikishena, I.S.	inhibition	ERP-N1, ERP-P2, late ERP	Global QATQS Rating: strong (score = 2.00); No additional bio-markers; N = 86	1.14
Event-related potentials in attention-deficit/hyperactivity disorder of the predominantly inattentive type: an investigation of EEG-defined subtypes [112]	Brown, C.R.; Clarke, A.R.; Barry, R.J.; McCarthy, R.; Selikowitz, M.; Magee, C.	inhibition	ERP-N2, ERP-P3, ERP-N1, ERP-P1, ERP-P2	Global QATQS Rating: moderate (score = 1.80); Additional bio-markers; N = 81	0.17
Lateralized modulation of posterior alpha oscillations in children [113]	Vollebregt, M.A.; Zumer, J.M.; Ter Huurne, N.; Castricum, J.; Buitelaar, J.K.; Jensen, O.	inhibition	MI-alpha	Global QATQS Rating: moderate (score = 1.80); Additional bio-markers; N = 21	0.49
Posterior alpha oscillations reflect attentional problems in boys with attention deficit hyperactivity disorder [114]	Vollebregt, M.A.; Zumer, J.M.; Ter Huurne, N.; Buitelaar, J.K.; Jensen, O.	inhibition	MI-alpha	Global QATQS Rating: moderate (score = 1.40); Additional bio-markers; N = 26	1.04
Intact stimulus–response conflict processing in ADHD—multilevel evidence and theoretical implications [115]	Bluschke, A.; Mückschel, M.; Roessner, V.; Beste, C.	inhibition	ERP-N1, ERP-P3, ERP-N2, ERP-P1	Global QATQS Rating: strong (score = 2.00); No additional bio-markers; N = 69	0.36
Topographical analyses of attention disorders of childhood [116]	DeFrance, J.; Smith, S.; Schweitzer, F.; Ginsberg, L.; Sands, S.	inhibition	PSD-beta, PSD-alpha, PSD-theta, ERP-P3, ERP-P2, ERP-P5	Global QATQS Rating: strong (score = 1.80); Additional bio-markers; N = 71	0.53
Varying required effort during interference control in children with AD/HD: task performance and ERPs [117]	Johnstone, S.J.; Watt, A.J.; Dimoska, A.	inhibition	ERP-N2, ERP-P3, ERP-P4	Global QATQS Rating: weak (score = 1.20); Additional bio-markers; N = 52	0.69
Response inhibition and interference control in children with AD/HD: a visual ERP investigation [118]	Johnstone, S.J.; Barry, R.J.; Markovska, V.; Dimoska, A.; Clarke, A.R.	inhibition	ERP-N1, ERP-N2, ERP-P3, ERP-P2	Global QATQS Rating: strong (score = 1.83); Additional bio-markers; N = 40	1.58
A comparative study on the neurophysiological mechanisms underlying effects of methylphenidate and neurofeedback on inhibitory control in attention deficit hyperactivity disorder [119]	Bluschke, A.; Friedrich, J.; Schreiter, M.L.; Roessner, V.; Beste, C.	inhibition	ERP-N1, ERP-N2, ERP-P3, ERP-P2	Global QATQS Rating: strong (score = 2.00); No additional bio-markers; N = 20	1.30
A pilot study of combined working memory and inhibition training for children with AD/HD [120]	Johnstone, S.J.; Roodenrys, S.; Phillips, E.;Watt, A.J.; Mantz, S.	inhibition, working memory	ERP-N1, ERP-N2, ERP-P3	Global QATQS Rating: weak (score = 1.17); Additional bio-markers; N = 40	1.91
Abnormal alpha modulation in response to human eye gaze predicts inattention severity in children with ADHD [121]	Guo, J.; Luo, X.; Wang, E.; Li, B.; Chang, Q.; Sun, L.; Song, Y.	inhibition	ERP	Global QATQS Rating: moderate (score = 1.60); No additional bio-markers; N = 108	0.13
Aiding diagnosis of childhood attention-deficit/hyperactivity disorder of the inattentive presentation: discriminant function analysis of multi-domain measures including EEG [122]	Johnstone, S.J.; Parrish, L.; Jiang, H.; Zhang, D.W.; Williams, V.; Li, S.	inhibition	PSD-alpha, PSD-beta, PSD-theta, PSD-delta, PSD-TBR	Global QATQS Rating: strong (score = 2.00); No additional bio-markers; N = 214	0.00
Behavioural and ERP indices of response inhibition during a stop-signal task in children with two subtypes of attention-deficit hyperactivity Disorder [123]	Johnstone, S.J.; Barry, R.J.; Clarke, A.R.	inhibition	ERP-N1, ERP-N2, ERP-P3, ERP-P2	Global QATQS Rating: moderate (score = 1.80); No additional bio-markers; N = 38	0.79
Virtual reality therapy in prolonging attention spans for ADHD [124]	Sushmitha, S.; Devi, B.T.; Mahesh, V.; Geethanjali, B.; Kumar, K.A.; Pavithran, P.	inhibition, planning	PSD-alpha, PSD-beta, PSD-theta, PSD-delta, PSD-TBR	Global QATQS Rating: strong (score = 2.33); Additional bio-markers; N = 20	0.00
Quantifying brain activity state: EEG analysis of background music in a serious game on attention of children [125]	Soysal, Ö.M.; Kiran, F.; Chen, J.	inhibition, planning	PSD-alpha, PSD-beta	Global QATQS Rating: strong (score = 2.50); No additional bio-markers; N = 6	0.79
Source-based multifractal detrended fluctuation analysis for discrimination of ADHD children in a time reproduction paradigm [126]	Khoshnoud, S.; Nazari, M.A.; Shamsi, M.	inhibition	ERP-P3- multifractality	Global QATQS Rating: moderate (score = 2.00); Additional bio-markers; N = 34	0.00
Virtual classroom: an ADHD assessment and diagnosis system based on virtual reality [127]	Tan, Y.; Zhu, D.; Gao, H.; Lin, T.W.; Wu, H.K.; Yeh, S.C.; Hsu, T.Y.	working memory	PSD-TBR, PSD-alpha, PSD-beta, PSD-theta, PSD-deta	Global QATQS Rating: weak (score = 1.20); Additional bio-markers; N = 100	1.45
Acquisition and analysis of cognitive evoked potentials using an emotiv headset for ADHD evaluation in children [128]	Mercado-Aguirre, I.M.; Gutiérrez-Ruiz, K.; Contreras-Ortiz, S.H.	working memory	ERP-P3	Global QATQS Rating: strong (score = 2.83); No additional bio-markers; N = 19	0.50
Personalized features for attention detection in children with attention deficit hyperactivity disorder [129]	Fahimi, F.; Guan, C.; Goh, W.B.; Ang, K.K.; Lim, C.G.; Lee, T.S.	inhibition	PSD, PSD-TAR, PSD-TBAR	Global QATQS Rating: strong (score = 2.33); No additional bio-markers; N = 120	1.19
Identification of ADHD cognitive pattern disturbances using EEG and wavelets Analysis [130]	Gabriel, R.; Spindola, M.M.; Mesquita, A.; Neto, A.Z.	inhibition, working memory	Morlet Wavelet Transform - delta, theta, alpha, and beta-power spectrum and amplitude	Global QATQS Rating: strong (score = 2.20); No additional bio-markers; N = 19	0.00
Neurofeedback based attention training for children with ADHD [131]	Chen, C.L.; Tang, Y.W.; Zhang, N.Q.; Shin, J.	inhibition, working memory	PSD-alpha, PSD-beta, PSD-theta, PSD-deta	Global QATQS Rating: strong (score = 2.50); Additional bio-markers; N = 10	n.a.
Influence of a BCI neurofeedback videogame in children with ADHD. Quantifying the brain activity through an EEG signal processing dedicated toolbox [132]	Blandón, D.Z.; Muñoz, J.E.; Lopez, D.S.; Gallo, O.H.	inhibition	PSD-alpha, PSD-beta, PSD-delta, PSD-gamma, PSD-TBR	Global QATQS Rating: strong (score = 2.83); No additional bio-markers; N = 9	2.64
Neurofeedback treatment experimental study for adhd by using the brain–computer interface neurofeedback system [133]	Liu, T.; Wang, J.; Chen, Y.; Wang, R.; Song, M.	inhibition	PSD-beta, PSD-theta, PSD-TBR, PSD- SMR	Global QATQS Rating: moderate (score = 1.83); Additional bio-markers; N = 22	2.63
Analysis of attention deficit hyperactivity disorder in EEG using wavelet transform and self organizing maps [134]	Lee, S.H.; Abibullaev, B.; Kang, W.S.; Shin, Y.; An, J.	working memory	Wavelet Transform-alpha, theta, beta power spectrum	Global QATQS Rating: strong (score = 2.60); No additional bio-markers; N = 39	1.09
Classification of ADHD and non-ADHD using AR models [135]	Marcano, J.L.L.; Bell, M.A.; Beex, A.L.	inhibition	PSD-TBR	Global QATQS Rating: weak (score = 3.00); No additional bio-markers; N = 4	1.19
Deep learning based on event-related EEG differentiates children with ADHD from healthy controls [136]	Vahid, A.; Bluschke, A.; Roessner, V.; Stober, S.; Beste, C.	inhibition	ERP-P3	Global QATQS Rating: strong (score = 1.20); Additional bio-markers; N = 28	3.44

**Table 3 sensors-22-04934-t003:** The five top scored articles according to the field-weighted citation impact metric by Scopus (the higher, the best). For each article, the normalized QATQS Score is also reported (the lower, the better). The normalized QATQS Score is computed as the ratio between the global quality score of the article and the number of quality components considered. FWCI: field-weighted citation impact.

Article	FWCI	Normalized QATQS Score
	Range [0.00–4.17]; Median = 0.77	Range [1.17–3.00]; Median = 2.17
[39]	4.17	1.83
[92]	4.16	2.00
[93]	3.66	2.00
[136]	3.44	1.20
[64]	3.25	2.20

**Table 4 sensors-22-04934-t004:** The five top scored articles according to the normalized QATQS Score. For each article, the field-weighted citation impact metric by Scopus is also reported (the higher, the best). The normalized QATQS score is computed as the ratio between the global quality score of the article and the number of quality components considered (the lower, the better). FWCI: field-weighted citation impact.

Article	Normalized QATQS Score	FWCI
	Range [1.17–3.00]; Median = 2.17	Range [0.00–4.17]; Median = 0.77
[120]	1.17	1.91
[127]	1.20	1.45
[136]	1.20	3.44
[117]	1.20	0.69
[89]	1.60	1.20

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
