# Peer review of "A Systematic Review on Feature Extraction in Electroencephalography-Based Diagnostics and Therapy in Attention Deficit Hyperactivity Disorder"

_sensors, 2022, doi:10.3390/s22134934_

Round 1
Reviewer 1 Report
This paper presents a systematic review on feature extraction strategies in an electroencephalographic (EEG) analysis of the executive functions of Attention Deficit Hyperactivity Disorder (ADHD) patients. The reference is suffiecent. However, some issues still remains.
1. The paper is not easy to follow. Please carefully polish the writing.
2. I recommend to add a timeline for some milestones. This will help readers understand the development of the ares comprehensively.
3. In Table 2, it is necessary to add the name of all mentioned papers.
4. There are so few explanation and disscution from Figure 4 to 10.
5. No conclusion?
6. Some neurocognitive related methods are missing, [1]Graph-based few-shot learning with transformed feature propagation and optimal class allocation, [2] Deep-irtarget: An automatic target detector in infrared imagery using dual-domain feature extraction and allocation.
7. There are many typos, such as "This function is usually tested in a oddball framework", "Author Contributions: da modificare!!! f".
Author Response
Please see the Section "Reviewer 1" in the attachment.

Reviewer 2 Report
The manuscript by Arpaia et al is written in a very confusing way. Title does not match the text, and introduction/discussion are very confusing and do not really correspond to the methods and results. Even more, language used is very complicated and would highly benefit from language editing,
Below I list problem that I have spotted:
Abstract is fragmented and not consistent
The description of diagnostical conditions in the Introduction is not that interesting in the context of this paper; biological basis is much more relevant – observed changes in ADHD should be introduced and it should be explained why EEG is a good tool to be used in that context.
Again presentation of executive functions is confusing. Please add the ADHD context and further explain how cognitive functions can be seen in EEG - it is more relevant that a summary of DSM5 or psychology textbook.
Questions to be addressed are far too ambitious and do not correspond to the work conducted.
Method description is again confusing – it is the first time where EF meets EEG, and then the description of EF is not an appropriate place in the "study labeling" subsection. Why it is important to label this functions?
Why Scopus index is used?
I do not understand why quality assessment is SO important for overview of EEG features being studied in the context of ADHD that it takes 0.5 of results and dicsussion...In results: what does it mean 10% supported the subjective..? why it should affect the quality of the paper?
Results:
Table 2 : all abbreviations should be explained. And authors’ names should be visible
Figs 4 – 10: what is not effective and effective??? How it is estimated?
I strongly advise to make emphasis on EEG features and not EFs…
The whole discussion is coming out of nowhere. Neither in the methods nor in the results section is anyone talking about findings of the included papers but the findings are discussed…
Author Response
Please see the Section "Reviewer 2" of the attachment.

Round 2
Reviewer 1 Report
Accept
Author Response
Please see Section Reviewer 1 of the attachment.

Reviewer 2 Report
I thank authors for the attempt to improve the manuscript. I still believe that very careful language editing is necessary as there are many misleading formulations, f.e. "With respect to high-level FEs, only planning is slightly more than the unit"
If authors insist on focusing of evaluation of quality of the papers, they should provide detailed scores on this evaluation an explicitly discuss what is wrong, what is mostly missing - this will help the reader, future authors and the cited authors to improve. Is it really so important to created TOP-5? What does it reflect in the overall context?
Inclusion criteria description: explanation should be consistently attached to inclusion criteria. No it is all over the place, including Results section
It should be also discussed what are the physiological/psychological backgrounds of the most studied components, and how particularly that relates to ADHD. It is extremely important in the context of "effective/ non effective" division that is not that clearly explained.
What are the arrows in tables near QATQS scores and Scopus index?
Author Response
Please see Section Reviewer 2 of the attachment.
